# The C-Terminal Repeat Units of SpaA Mediate Adhesion of *Erysipelothrix rhusiopathiae* to Host Cells and Regulate Its Virulence

**DOI:** 10.3390/biology11071010

**Published:** 2022-07-05

**Authors:** Chao Wu, Zhewen Zhang, Chao Kang, Qiang Zhang, Weifeng Zhu, Yadong Zhang, Hao Zhang, Jingfa Xiao, Meilin Jin

**Affiliations:** 1State Key Laboratory of Agricultural Microbiology, Huazhong Agricultural University, Wuhan 430070, China; wudlnchao123@163.com (C.W.); kangchaosuper@163.com (C.K.); zhuweifeng8@126.com (W.Z.); 2College of Veterinary Medicine, Huazhong Agricultural University, Wuhan 430070, China; 3National Genomics Data Center, Beijing Institute of Genomics, Chinese Academy of Sciences and China National Center for Bioinformation, Beijing 100101, China; zhangzw@big.ac.cn (Z.Z.); zhangyadong@big.ac.cn (Y.Z.); zhanghao17m@big.ac.cn (H.Z.); 4CAS Key Laboratory of Genome Sciences and Information, Beijing Institute of Genomics, Chinese Academy of Sciences and China National Center for Bioinformation, Beijing 100101, China; 5College of Biomedicine and Health, Huazhong Agricultural University, Wuhan 430070, China; zhangq_0401@mail.hzau.edu.cn; 6University of Chinese Academy of Sciences, Beijing 100049, China

**Keywords:** *Erysipelothrix rhusiopathiae*, comparative genomics, virulence, SpaA, C-terminal repeat

## Abstract

**Simple Summary:**

*Erysipelothrix rhusiopathiae* is an important zoonotic pathogen, which poses a serious harm to the pig industry. We aimed to evaluate the genomic differences between virulent and avirulent strains to study the pathogenic mechanism of *Erysipelothrix rhusiopathiae*. The results showed that the *spaA* gene of avirulent strain lacked 120bp, encoding repeat units at the C-terminal of SpaA, the virulence of the virulent strain with this 120 bp deletion was attenuated, and the mutant strain decreased adhesion to porcine iliac artery endothelial cells.

**Abstract:**

*Erysipelothrix rhusiopathiae* is a causative agent of erysipelas in animals and erysipeloid in humans. However, current information regarding *E. rhusiopathiae* pathogenesis remains limited. Previously, we identified two *E. rhusiopathiae* strains, SE38 and G4T10, which were virulent and avirulent in pigs, respectively. Here, to further study the pathogenic mechanism of *E. rhusiopathiae*, we sequenced and assembled the genomes of strains SE38 and G4T10, and performed a comparative genomic analysis to identify differences or mutations in virulence-associated genes. Next, we comparatively analyzed 25 *E. rhusiopathiae* virulence-associated genes in SE38 and G4T10. Compared with that of SE38, the *spaA* gene of the G4T10 strain lacked 120 bp, encoding repeat units at the C-terminal of SpaA. To examine whether these deletions or splits influence *E. rhusiopathiae* virulence, these 120 bp were successfully deleted from the *spaA* gene in strain SE38 by homologous recombination. The mutant strain *ΔspaA* displayed attenuated virulence in mice and decreased adhesion to porcine iliac artery endothelial cells, which was also observed using the corresponding mutant protein SpaA’. Our results demonstrate that SpaA-mediated adhesion between *E. rhusiopathiae* and host cells is dependent on its C-terminal repeat units.

## 1. Introduction

*Erysipelothrix* are Gram-positive, rod-shaped, facultative anaerobic bacteria that cause diseases in a variety of animals including swine, humans, poultry, sheep, cattle, and wild animals [1]. Twenty-five serovars have been described for the genus *Erysipelothrix* [2]. In current taxonomy, this genus contains two main species, *E. rhusiopathiae* and *E. tonsillarum*, each consisting of various serotypes, including serotypes 1a, 1b, 2, 4, 5, 6, 8, 9, 10, 11, 12, 15, 16, 17, 19, 21, and N in *E. rhusiopathiae*, and serotypes 3, 7, 10, 14, 20, 22, and 23 in *E. tonsillarum* [3]. Moreover, serovars 13 and 18 are considered to be members of two separate and new species [2]. Based on DNA–DNA hybridization analysis of numerous isolates from different organisms, *E. rhusiopathiae* is considered to be the pathogenic species of the genus [3], causing swine erysipelas with symptoms ranging from acute septicemia to chronic endocarditis and polyarthritis [2]. Swine erysipelas occurs worldwide and is of economic importance throughout Europe, Asia, Australia, and the Americas [4]. In China, the occurrence of swine erysipelas has significantly decreased since the 1990s because of increased and improved management of the industry [1]. However, since 2012, the incidence of swine erysipelas has emerged at an alarming rate, developing from scattered occurrences in a small number of farms to systemic outbreaks in many provinces, thereby causing large financial losses [5]. In addition, *E. rhusiopathiae* is also a zoonotic pathogen that causes erysipeloid in humans [6]. Moreover, this pathogen can transfer from diseased pigs to humans [7], which undoubtedly increases the risk of this disease in humans, especially in individuals working with pigs or pork-derived products.

Previous studies identified several candidate virulence factors of *E. rhusiopathiae* such as neuraminidase, capsular antigens, rhusiopathiae surface protein A (*rspA*), rhusiopathiae surface protein B (*rspB*), substrate-binding protein (*sbp*), surface protective antigen (*spaA*), and the 64–66 kDa antigen; however, few of these candidate virulence factors have been demonstrated to play a critical role in pathogenesis [6]. The role of SpaA in virulence is the only one to have been verified in mice; it acts as an adhesin between *E. rhusiopathiae* and host cells [3]. Therefore, information regarding the pathogenic mechanism underlying *E. rhusiopathiae* infection remains limited [8]. This knowledge gap has become one of the most important obstacles to controlling the infection. Thus, the identification of novel virulence factors is necessary to improve our understanding of *E. rhusiopathiae* pathogenesis [5].

“Omics” study is an important strategy to screen virulence-related factors, including genomics, proteomics, and transcriptomics [9]. Previously, we isolated and identified two strains, SE38 and G4T10; strain SE38 is virulent in pigs, whereas strain G4T10 is avirulent. The characteristics of the porcine heart host responses to these two strains were also analyzed by transcriptomics. The results showed that strain SE38 could significantly induce inflammatory and immune responses, whereas strain G4T10 could not [10]. In this study, to elucidate the mechanism underlying the differential toxicity of the virulent strain SE38 and avirulent strain G4T10, genome sequencing and assembly of these two strains were conducted. Furthermore, putative virulence-related genes or sequences were screened by comparative analysis and verified by animal infection with a deletion mutant.

## 2. Material and Methods

### 2.1. Bacterial Strains and Cultivation

The bacterial strains and plasmids used in this study are listed in Appendix A. *E. rhusiopathiae* G4T10 is an avirulent strain, and *E. rhusiopathiae* SE38 is a virulent strain. SE38, G4T10, and mutant strain *ΔspaA* were cultured in tryptic soy broth (TSB) liquid or agar medium (TSA) (Difco, Detroit, MI, USA) supplemented with 10% bovine sera at 37 °C. *Escherichia coli* DH5a and BL21(DE3) were cultured in Luria Bertani (LB) liquid medium or plated on LB agar at 37 °C.

### 2.2. Genome Sequencing and Assembly

The genomic DNA of strains G4T10 and SE38 was extracted using a DNA extraction kit (TaKaRa DNAiso, reagent code: D305A; TaKaRa Biotechnology Co., Ltd., Dalian, China). Strains G4T10 and SE38 were sequenced using PacBio RS II (Pacific Biosciences, San Francisco, CA, USA). SMRT cell sequencing generated 68,227- and 58,600-long reads for G4T10 and SE38, respectively. The sums of bases were 455,449,435 and 436,809,021, and the average read lengths were up to 6675 bp and 7454 bp, respectively. PacBio RS II yielded 190-fold read coverage in strain G4T10, and approximately 182-fold read coverage in strain SE38. All PacBio reads of each strain were assembled into one polished contig by HGAP v3 [11], and each contig of the strain constituted one complete circular genome.

### 2.3. Genome Annotation and Analysis

The tRNA and rRNA genes were identified by tRNAscan-SE [12] and RNAmmer [13], respectively, and the coding sequences (CDS) were predicted with Prodigal, version 2.60 [14]. The predicted genes were annotated by performing a BLAST [15] search against databases of non-redundant proteins from the NCBI and COG (E-value 1 × 10^−5^). Virulence gene predictions were made by performing a BLAST search against the virulence factor database (VFDB) [16]. The virulence genes were compared using R package genoPlotR [17]. Sequence alignment was performed by MAFFT and displayed using TEXshade [18].

### 2.4. Pan-Genomic Analyses and Genome Comparison

The gene clusters of the six strains were identified by PGAP [19] using the MP method with a 50% cut-off for protein sequence identity and an E-value cut-off of 1 × 10^−5^. Pan-genome characteristic curves were depicted by PanGP [20] with DG sampling algorithms. The multiple-genome alignment of the six complete and draft strains was performed using Mauve [21].

### 2.5. Phylogenetic Analyses

Phylogenetic trees were constructed based on the 16S rRNA and single-copy core genes, with *Staphylococcus aureus* NCTC8325 as the out-group. The software MAFFT was used to align each cluster of the single-copy core gene sequences from the six strains. The recombination genes in these clusters were detected using PhiPack [22] (which contains three individual methods for calculating *p*-values, namely, the neighbor similarity score (NSS), Maxχ^2^, and Phi) and GENECONV [23] with *p*-values < 0.05. We removed 2.5% of the single-copy gene clusters (16 clusters) that showed evidence of recombination by all four methods. Next, 613 clusters were aligned and concatenated in order to construct a rooted maximum likelihood (ML) species tree using PhyML v3.0 [24] with 100 bootstrap replicates. The rooted species tree based on 16S rRNA was also constructed using PhyML v3.0 with 100 bootstrap replicates.

### 2.6. Gene Gain and Loss Analyses

Gene gain/loss evolutionary events, including gene loss, gene duplication, speciation, lateral gene transfer (LGT), and gene birth or genesis, were inferred using the AnGST program. The most parsimonious reconciliation was obtained by inferring a minimum set of evolutionary events. Gene trees were constructed for every gene cluster containing three or more sequences using PhyML v3.0. Reconciliation of each gene tree and the species tree was performed using the default event penalty values of AnGST (LGT = 3.0, duplication = 2.0, loss = 1.0, and speciation = 0). Gene/species tree reconciliation can recover the optimal set of evolutionary events that explain any topological discordance between the gene and species trees. A birth event in the genome was defined as the presence of only one gene in the cluster. The method described by Richards et al. [25] was used to explain the evolutionary history of gene clusters containing two genes.

### 2.7. RNA-Seq Library Construction and Sequencing

Total RNA was extracted from strains G4T10 and SE38 using the Trizol method. Ribosomal RNA was removed using the Epicentre Ribo-Zero rRNA Removal Kit (Gram-Negative Bacteria) following the manufacturer’s instructions. We constructed strand-specific cDNA libraries (two biological replicates for each strain) using the KAPA Stranded mRNA-Seq Kit Illumina platform (KAPA BIOSYSTEMS, dUTP Based, Wilmington, MA, USA) according to the manufacturer’s instructions. Paired-end libraries of strains G4T10 and SE38 with insert sizes of 300–400 bp were sequenced using Illumina HiSeq2000 with a sequencing read length of 101 bp.

### 2.8. Read Mapping and Gene Expression Analysis

The sequencing data were quality pre-processed using a Perl script. Adaptor reads, reads with >2% ‘N’ bases, and reads with low quality (<15) for over half of their lengths were filtered out. High-quality reads were mapped to the genome sequences of strain G4T10 and SE38 using Bowtie version 0.12.7 [26]. Reads that mapped to rRNA genes or that failed to align to genome sequences were removed. Next, we focused on transcripts that mapped to the core genes of the two strains. The script in HTSeq [27] was used to count the reads aligned to each gene. These results were used to identify differentially regulated genes using the DEGseq2 [28] package with *q*-value ≤ 0.05 and log2 fold-change > 1. The operon organization of *spaA* was determined using Rockhopper [29].

### 2.9. Construction of the ΔSpaA Mutant

The deleted mutant strain was constructed as previously described [5]. DNA fragments of the *spaA* gene were amplified from G4T10 genomic DNA using primers *ΔspaA*-F and *ΔspaA*-R listed in Appendix A, and then inserted into pSET4s to construct knockout plasmid pSET4s-*ΔspaA*. Next, the pSET4s-*ΔspaA* plasmid was electrotransformed into SE38 competent cells. The mutant strain was screened based on thermosensitive suicide and spectinomycin resistance encoded by the pSET4s vector, and then verified using two pairs of primers: P1/P2 (to detect the *spaA* gene) and P3/P4 (to detect the 16S rRNA gene).

### 2.10. Bacterial Cell Adhesion Assay

Bacteria were harvested, washed, and then resuspended in PBS containing 10 μM of carboxyfluorescein diacetate succinimidyl ester (CFDA-SE). The bacteria were incubated with gentle rotation at 37 °C for 30 min in the dark. Porcine iliac artery endothelial cells (PIECs) were cultured in a 12-well plate at 37 °C and 5% CO_2_ in RPMI 1640 medium (Gibco). Labeled bacteria and PIECs were co-incubated (100:1 ratio) at 37 °C for 2 h. The infected cells were washed three times with PBS, then harvested with 0.25% trypsin and fixed with 4% paraformaldehyde. PIECs incubated with unlabeled bacteria served as negative controls. Flow cytometry data were analyzed using FlowJo software, with a total of 10,000 cells counted in each experiment.

### 2.11. Evaluation of ΔSpaA Pathogenicity in Mice

To compare the pathogenicity of *ΔspaA*, WT strain SE38, and attenuated strain G4T10, 40 C57BL/6 mice (female, 4 weeks old) were randomly divided into four groups for an infection experiment. The mice were subcutaneously challenged with 500 μL of bacterial suspension or PBS. Clinical signs and mortality were then observed for 10 days.

### 2.12. Expression of Recombinant Proteins

The entire open reading frame of the *spaA* gene was amplified from SE38 or *ΔspaA* with primers SpaA-F and SpaA-R (Appendix A), and then inserted into the pET28a vector. Next, the recombinant vectors were transformed into *E. coli* BL21(DE3) for expression of SpaA (from SE38) and SpaA’ (from *ΔspaA*). Recombinant proteins were purified using nickel affinity chromatography (Novagen).

### 2.13. Analysis of Recombinant Protein Adhesion to PIECs

Cells were cultured in a 6-well plate (1 × 10^6^ cells/well), and incubated with 20 µg/mL of recombinant protein SpaA or SpaA’ for 30 min. The cells were washed three times with PBS and fixed in 4% paraformaldehyde, then again washed three times, blocked with 1% BSA in PBS for 30 min, and incubated at room temperature with a His-Tag monoclonal antibody (1:1000 dilution) in 1% BSA-PBS for 1 h. Flow cytometry was performed using a FITC-conjugated goat anti-mouse IgG, with a total of 10,000 cells counted in each experiment.

### 2.14. Data Accessibility

The two genome sequences of *E. rhusiopathiae* G4T10 and *E. rhusiopathiae* SE38 have been submitted to the NCBI under accession numbers SAMN03770844 and SAMN03770845. The corresponding transcriptome data have been deposited in the Genome Sequence Archive (GSA) [30] in National Genomics Data Center [31], Beijing Institute of Genomics (BIG), Chinese Academy of Sciences and China National Center for Bioinformation, under accession number CRA000357, which are publicly accessible at https://ngdc.cncb.ac.cn/gsa/browse/CRA000357 (accessed on 15 July 2021).

## 3. Results

### 3.1. General Genomic Features of Strains G4T10 and SE38

The general genome features of *E. rhusiopathiae* G4T10, *E. rhusiopathiae* SE38, and four previously sequenced *E. rhusiopathiae* strains are summarized in Appendix A. *E. rhusiopathiae* G4T10 is the first sequenced avirulent strain of the class Erysipelotrichia. *E. rhusiopathiae* G4T10 has one complete circular genome, 1,770,505 bp in size, with a GC content of 36.5%. The complete genome of *E. rhusiopathiae* SE38 is 7.6 kb larger than strain G4T10 (total size 1,778,153 bp) and has the same GC content. *E. rhusiopathiae* G4T10 has 1681 predicted CDS, 4 copies of the 16S-23S-5S rRNA operon, and 54 tRNA genes. The *E. rhusiopathiae* SE38 genome is predicted to have 1683 CDS, 5 copies of the 16S-23S-5S rRNA operon, and 55 tRNA genes (Appendix A). The genomes of *E. rhusiopathiae* G4T10 and SE38 contain fewer rRNA operons than previously sequenced strains Fujisawa and GXBY-1, but more operons than strains SY1027 and ATCC19414.

### 3.2. Comparative Genomic Analysis Results

All 10,274 protein CDS in the six *E. rhusiopathiae* strains were clustered using PGAP. All protein CDS clustered into 2057 orthologs; 1530 (74.4%) orthologs were identified in the six strains as the core genome (Figure 1a), and 139 (6.8%) orthologs were identified as dispensable clusters (shared by two to five strains), and 324 genes were strain-specific (Appendix A). Strains Fujisawa, G4T10, and SE38 had fewer strain-specific genes, implying high genome similarity between them. However, strain SY1027 had the greatest number of strain-specific genes, indicating it varies the most from the other strain genomes. The pan-genome profile fitted curves of the six strains shows the relations between the pan-genome size, the core genome size, and the strain number (Figure 1b). As shown in Figure 1b, we can intuitively observe that the pan-genome becomes larger when a new strain is added (blue curve); however, the core genome size was relatively stable and converged rapidly (green curve), revealing that *E. rhusiopathiae* has an open pan-genome. The genome sequences of the six *E. rhusiopathiae* strains are highly conserved with no apparent large-scale rearrangement (Figure 1c). There are four long locally collinear blocks (LCBs) with a minimum weight of 174 in each genome. Strains G4T10 and SE38 are more closely related to strain Fujisawa than to the other strains.

To clarify the phylogenetic relationship of the six *E. rhusiopathiae* strains, we constructed phylogenetic trees based on concatenated single-copy core genes using *Staphylococcus aureus* NCTC8325 as the out-group (for details, see the Materials and Methods section; Figure 1d). According to this tree, the six *E. rhusiopathiae* strains clearly group into two different branches. Strains SE38, GXBY-1, SY1027, G4T10, and Fujisawa are tightly grouped, indicating that they are more closely related than strain ATCC19414. All strains except ATCC19414 were isolated in Asia; thus, the closer phylogenetic relationship may reflect their high genetic conservation. Strain ATCC19414 formed a separate clade in the tree; this divergence may be due to special distance constituting a strong physical barrier to gene exchange.

To obtain insights into gene family dynamics, we reconciled every gene tree with the species tree using AnGST [32]. Gene-tree species-tree reconciliation may help us to explicitly infer the evolutionary history of every gene family in the dataset [33]. The evolutionary history usually includes gene birth, duplication, loss, and horizontal gene transfer (HGT) events. Our results suggest that the last common ancestor of all the species had a genome containing 1260 genes (Figure 1e). Strain ATCC19414 has maintained all 1260 acquired genes, whereas strain SE38 has lost the greatest number of genes (1006 genes). The gene loss events that occurred in the group including strain SE38 and strain GXBY-1 (204 losses) outnumbered those in the strain SY1027, strain G4T10, and strain Fujisawa group (four losses) during their separation. Gene gains occurred at different times during *Erysipelothrix* species evolution. However, most horizontally acquired genes were lost during *Erysipelothrix* species evolution. These results suggest that gene loss was a major evolutionary driving force during the dynamic process of *Erysipelothrix* as intracellular bacteria. These results are similar to previous studies examining intracellular bacteria [34,35,36].

### 3.3. Potential Virulence Factors

All predicted genes were searched against the VFDB [16] using BLASTp to identify potential virulence factors. All virulence factors in VFDB are organized into four super-families including adhesion, secretion, iron, and toxin. Several candidate virulence factors were identified in *E. rhusiopathiae* strains based on VFDB (Appendix A). However, no toxins were detected in these strains, which is consistent with previous research [37]. The 25 identified virulence genes of *E. rhusiopathiae* were extracted and concatenated (Appendix A). No obvious virulence gene gain or loss was observed in the six *E. rhusiopathiae* strains.

The comparative results showed that three of these twenty-five virulence genes, *rspA*, *sbp*, and *spaA*, were obviously different (Figure 2a). Of these, both the *rspA* and *sbp* gene were split into two short genes in strain G4T10; however, the causes of the gene split were different. The split in the *rspA* gene was due to a single guanine deletion (Figure 2b), compared to a single base mutation (T→G) in the *sbp* gene (Figure 2c). The *spaA* gene lost 120 bases at its C-terminal in strain G4T10 (Figure 2d). The SpaA protein in all five highly virulent strains (*E. rhusiopathiae* SE38, Fujisawa, GXBY-1, SY1027, and ATCC19414) contained eight C-terminal repeat units (cell wall binding, CW_binding) consisting of 20 amino acids, which is consistent with results of previous studies [38,39]. Each of the repeats began with the dipeptide glycine–tryptophan (GW). The eight repeats were very similar, and the fourth, fifth, and seventh repeats were identical (Appendix A). However, we detected 40 amino acids’ deletion in the SpaA protein in attenuated strain G4T10, resulting in the absence of two C-terminal repeat units (the second and the third repeats in strain SE38) compared to the five highly virulent strains. The C-terminal repeat units of the SpaA protein are necessary for the tight binding of SpaA to the bacterial surface [38,39]. These repeat units can attach surface proteins to the cell wall through non-covalent interactions [40,41]. Although *spaA* was transcriptionally active in both G4T10 and SE38, *spaA* gene expression was different (FPKM = 19.33 in G4T10 and FPKM = 96.02 in SE38). The operon organization results showed that *spaA* is a monocistronic gene in both G4T10 and SE38, consistent with operon DB in strain Fujisawa [42].

### 3.4. Characteristics of the Mutant Strain

To evaluate the influence of the different *spaA* sequences in strains SE38 and G4T10 on *E. rhusiopathiae* virulence, a deletion mutant of the differing sequence (*ΔspaA*) was constructed in strain SE38 by homologous recombination; the mutation was confirmed by PCR with two primer pairs (P1/P2, P3/P4). The results showed that the molecular weight of the *spaA* gene in the SE38 strain was higher than in the G4T10 and *ΔspaA* strains (Figure 3A). Sequencing analysis further indicated that mutant strain *ΔspaA* had been successfully constructed. As a control, the 16S RNA gene could be detected in SE38, G4T10, and *ΔspaA*. To examine the characteristics of the mutant strain, growth curves and transmission electron microscopy of *ΔspaA*, SE38, and G4T10 were conducted (Figure 3B,C). The results indicated that mutation of the *spaA* gene does not influence the growth and capsule of *E. rhusiopathiae*.

### 3.5. ΔSpaA Displays Attenuated Virulence in Mice, and Decreased Adhesion to PIECs

To examine whether mutation of the *spaA* gene influenced *E. rhusiopathiae* virulence, an animal experiment was performed using C57BL/6 mice. The survival rate of mice in the *ΔspaA* group until 10 days post-infection was 70%, but only 10% in the SE38 group; all mice in the G4T10 group survived without any obvious symptoms (Figure 4a). These results indicate that mutation of the *spaA* gene significantly attenuated *E. rhusiopathiae* virulence in mice.

A previous study demonstrated that SpaA influences *E. rhusiopathiae* virulence mainly via its adhesion to host cells [3,4]. Based on this, we hypothesized that the *ΔspaA* strain displayed attenuated virulence perhaps because the mutation of the *spaA* gene influenced the adhesive ability of SpaA. To confirm this hypothesis, the adhesive ability of the *ΔspaA* strain was tested in PIECs by flow cytometry. The adhesive ability of the *ΔspaA* mutant to PIECs was obviously decreased compared with that of wild-type strain SE38 (Figure 4b).

### 3.6. The Deleted Sequence Influences the Adhesive Ability of SpaA in PIECs

To further examine whether the deleted sequence influences the adhesive ability of SpaA, the *spaA* genes of SE38 and *ΔspaA* were cloned and expressed. The recombinant SpaA encoded by *ΔspaA* was named SpaA’. SDS-PAGE and Western blotting analyses showed that the molecular weight of SpaA’ was lower than that of wild-type SpaA (Figure 5a,b). Next, flow cytometry analysis was performed to determine the adhesive ability of SpaA and SpaA’ to PIECs. We observed that the adhesive ability of SpaA was significantly higher than that of SpaA’ (Figure 5c), indicating that the deleted sequence plays an important role in the adhesive ability of SpaA.

## 4. Discussion

*E. rhusiopathiae* is an opportunistic pathogen that can cause acute septicemia or chronic endocarditis and polyarthritis in pigs [1]. Since 2012, *E. rhusiopathiae* infection has recurred as a serious clinical problem in pig populations in China after 20 years of hibernation [10]. However, currently, knowledge of *E. rhusiopathiae* pathogenesis remains limited, which severely impedes the development of effective vaccines or drugs. Here, to investigate the pathogenic mechanism of *E. rhusiopathiae*, a comparative genomic analysis was performed for six *E. rhusiopathiae* strains, including the avirulent strain G4T10 and several virulent strains (SE38, Fujisawa, SY1027, GXBY-1, and ATCC19414). The avirulent strain G4T10 and virulent strain SE38 have a striking difference in their virulence phenotypes. Bacterial vaccines play an important role in the prevention and control of swine bacterial diseases. Since the G4T10 strain is avirulent, it has the potential to be a live vaccine compared to virulent strains such as SE38 or Fujisawa. Initially, we hypothesized that this was due to differences in their genome structures; however, the *E. rhusiopathiae* strains were shown to be highly conserved, and no large-scale rearrangement was observed. Phylogenetic analysis showed that strains SE38, Fujisawa, SY1027, GXBY-1, and G4T10 have a closer phylogenetic relationship, which reflects the high genetic conservation of Asia-originating *E. rhusiopathiae* strains. Genome dynamics analysis results suggested that gene loss is a major evolutionary driving force during the dynamic evolution of *E. rhusiopathiae* as an intracellular bacterium. Virulence gene analysis indicated that the difference in virulence of *E. rhusiopathiae* G4T10 and that of other virulent *E. rhusiopathiae* strains is not due to gene gain or loss at the genomic level, but rather to a spilt or variation in some shared virulence genes. Interestingly, compared with the virulent SE38, the avirulent G4T10 has a 120bp deletion in *SpaA*. Therefore, the *SpaA* gene of a large number of virulent and avirulent strains needs to be analyzed later to determine whether this 120bp can be used as a marker to distinguish isolate of *E. rhusiopathiae* is virulent or not.

SpaA was previously identified as a major protective antigen of *E. rhusiopathiae* [43]. However, recent studies have shown that SpaA also plays an important role in *E. rhusiopathiae* virulence dependent on its adhesion to host cells [3,4,44]. Numerous studies have shown that SpaA can mediate the adhesion of *E. rhusiopathiae* to porcine endothelial cells via phosphorylcholine without platelet-activating factor receptor [45]; however, the key domain that influences the adhesive ability of SpaA was unclear. In this study, we demonstrated that the repeat sequence of the C-terminal repeat units obviously affects the adhesive ability of SpaA, which then attenuates *E. rhusiopathiae* virulence. These results further elucidate the molecular mechanism underlying SpaA adhesion.

Our previous research demonstrated obvious differences in virulence of strains SE38 and G4T10 [10]. However, in this study, comparative analysis indicated that there are only a few differences between the genomes of these strains. Of these, the different virulence-associated genes include *spaA*, *sbp*, and *rspA*, which were split in two in strain G4T10. In addition to *spaA*, which was examined in this study, *sbp* has also been shown to be associated with virulence, based on pig survival experiments with deletion mutant strains [5]. In addition, several studies have indicated that RspA expressed by the cloned *rspA* gene may influence *E. rhusiopathiae* virulence via participation in biofilm formation [2]. Unfortunately, attempts to construct a mutant strain to examine the influence of the split *rspA* gene on *E. rhusiopathiae* virulence were unsuccessful (data not shown). Thus, the effect of *rspA* on *E. rhusiopathiae* virulence remains to be verified. Taken together, our results explain why the *ΔspaA* mutant displayed attenuated virulence in mice compared with wild-type strain SE38, but higher virulence than strain G4T10.

## 5. Conclusions

Our data showed that the genomes of *E. rhusiopathiae* strains G4T10 and SE38 were highly similar, and that the difference in their virulence was mainly due to the mutation of some virulence-related genes, especially *spaA*. We demonstrated that the adhesion ability of SpaA depends on the repeat units at its C-terminus, which significantly influence *E. rhusiopathiae* virulence. These results further our understanding of *E. rhusiopathiae* pathogenesis.

## Figures and Tables

**Figure 1 biology-11-01010-f001:**
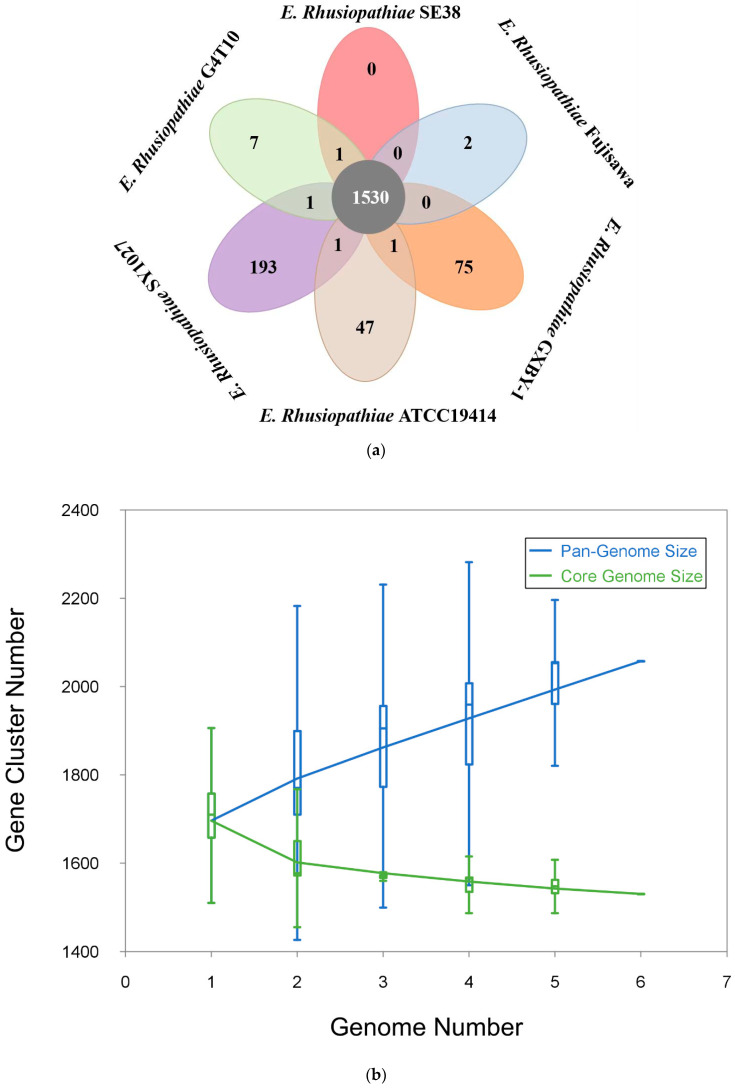
Pan-genome analyses of *E. rhusiopathiae*. (**a**) Flower plots showing the core gene number (in the center) and strain-specific gene number (in the petals) of the six *E. rhusiopathiae* strains. (**b**) Pan-genome and core genome curves of the six *E. rhusiopathiae* strains. (**c**) Mauve alignment of the six *E. rhusiopathiae* strains. (**d**) Phylogenetic tree of *E. rhusiopathiae* strains based on the 16S rRNA gene. (**e**) Phylogenetic tree of *E. rhusiopathiae* strains based on the concatenated alignment of single-copy core genes. The numbers in boxes on every node represent the genome size and the overall balance (gain–loss: balance), with black indicating the genome size, red indicating an overall gain, and blue indicating an overall loss. The numbers above and below every lineage show the number of birth/duplication/transfer and loss events.

**Figure 2 biology-11-01010-f002:**
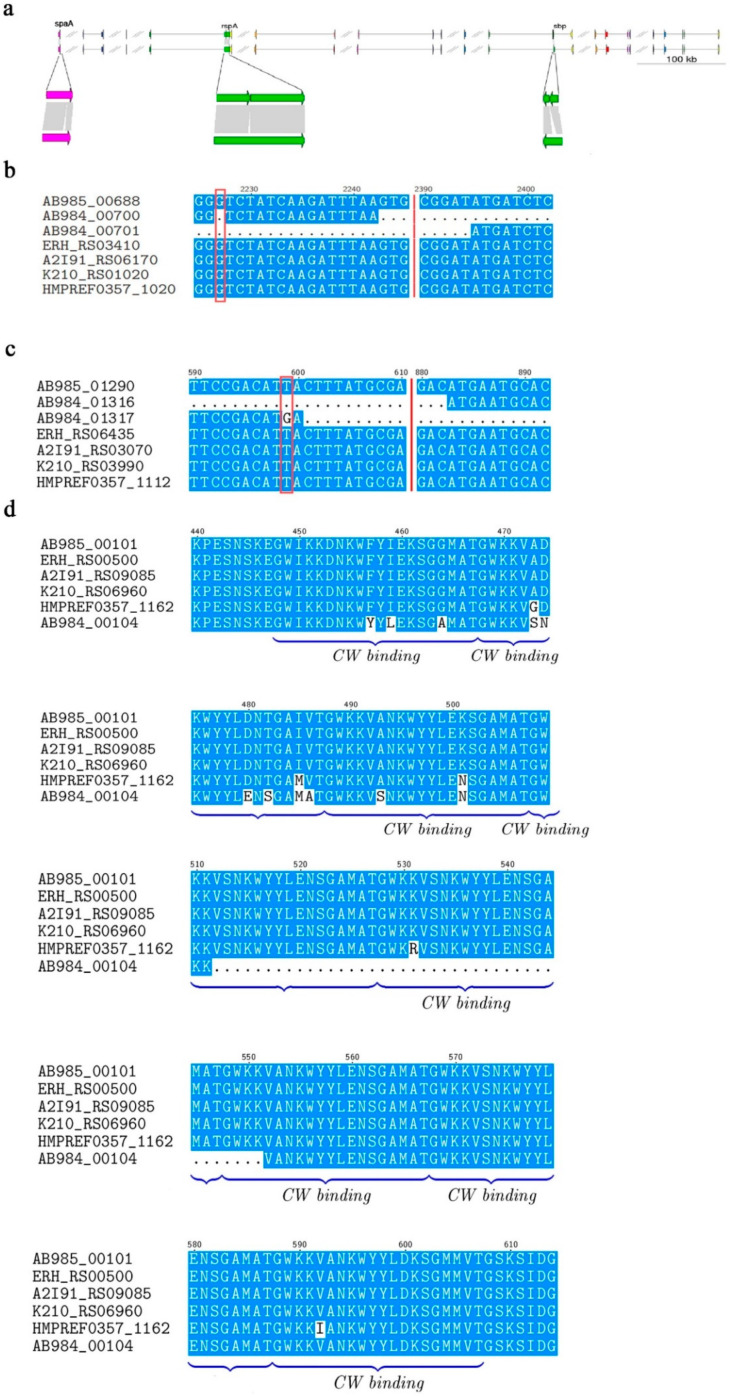
Virulence gene comparison. (**a**) Comparison of 25 identified virulence genes in *E. rhusiopathiae* G4T10 and SE38. (**b**) Partial nucleotide sequence alignment of the *rspA* genes from the six *E. rhusiopathiae* strains (the single guanine deletion is marked with a red box). (**c**) Partial nucleotide sequence alignment of the *sbp* genes from the six *E. rhusiopathiae* strains (the single base mutation is marked with a red box). (**d**) Partial amino acid sequence alignment of the SpaA protein from the six *E. rhusiopathiae* strains. The blue brackets represent the eight C-terminal repeat units (cell wall binding, CW_binding).

**Figure 3 biology-11-01010-f003:**
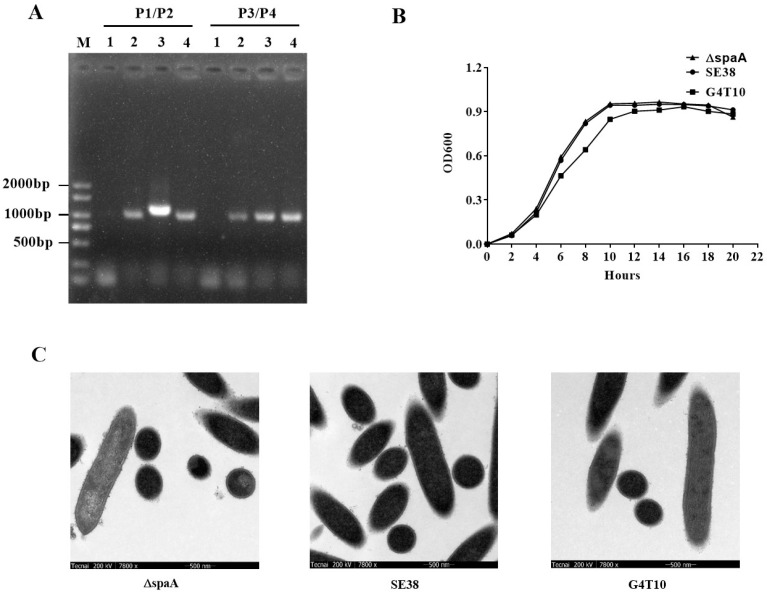
Construction and confirmation of the *ΔspaA* mutant. (**A**) Confirmation of the *ΔspaA* mutant by PCR using primer pairs P1/P2 (to detect the *spaA* gene) and P3/P4 (to detect the 16S rRNA gene). Lane 1, negative control; Lane 2, *ΔspaA*; Lane 3, SE38; Lane 4, G4T10. (**B**) Growth curves of *ΔspaA*, SE38, and G4T10. The bacteria were cultured in TSB supplemented with 10% bovine sera at 37 °C. Absorbance at 600 nm was measured at 2 h intervals. The results shown are representative of three independent experiments. (**C**) The capsules of *ΔspaA*, SE38, and G4T10 detected by transmission electron microscopy (×7800).

**Figure 4 biology-11-01010-f004:**
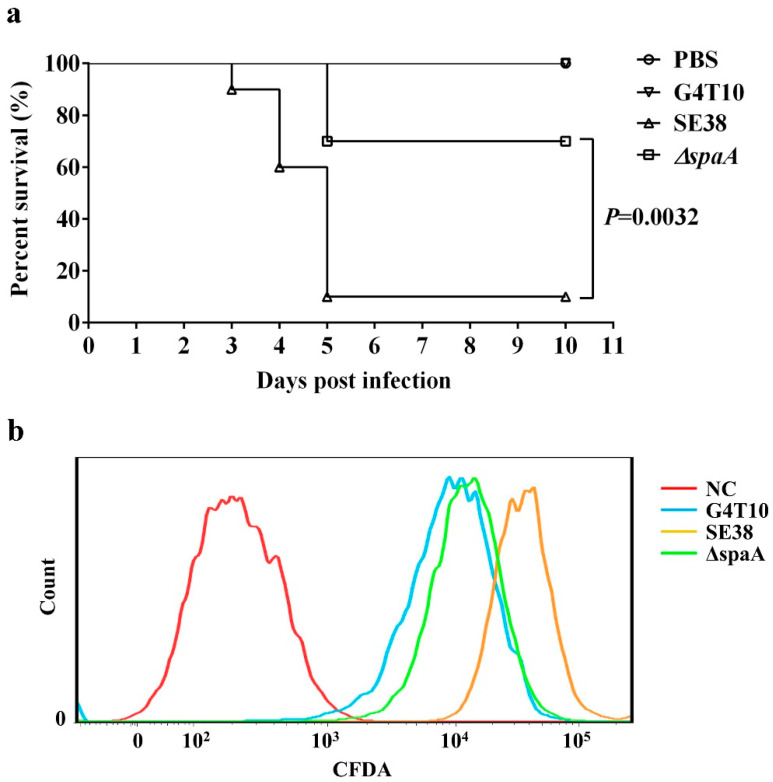
Evaluation of the virulence and adhesion ability of *ΔspaA*. (**a**) Mouse survival rate. C57BL/6 mice were subcutaneously challenged with 500 μL of attenuated strain G4T10, WT strain SE38, mutant strain *ΔspaA*, or PBS (10 mice/group). (**b**) Flow cytometry analysis. G4T10, SE38, and *ΔspaA* were labeled with CFDA-SE, and then co-incubated with PIECs (100:1 ratio) at 37 °C for 2 h. Flow cytometry was performed with a total of 10,000 cells counted in each group. Survival assay statistics were conducted using the log-rank (Mantel-Cox) test. All experiments were performed at least twice under similar conditions and yielded similar results.

**Figure 5 biology-11-01010-f005:**
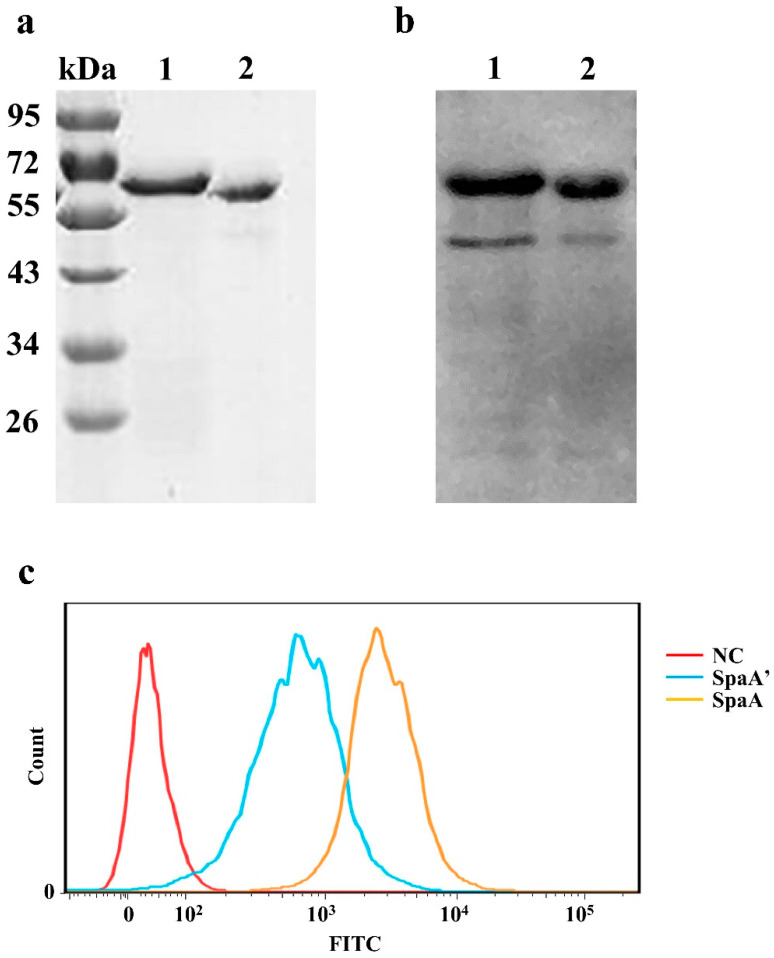
Adhesion ability analysis of mutated SpaA’. (**a**) SDS-PAGE analysis and (**b**) Western blot analysis of recombinant SpaA and SpaA’. The blot was probed with an anti-His-Tag monoclonal antibody (Cali-Bio). Lane 1: SpaA; Lane 2: SpaA’. (**c**) Flow cytometry analysis. PIECs were incubated with 20 µg/mL of SpaA or SpaA’ for 30 min, then washed three times with PBS and fixed in 4% paraformaldehyde. After another three washes, the cells were blocked with 1% BSA in PBS for 30 min, and then incubated with the His-Tag monoclonal antibody (1:1000 dilution) in 1% BSA-PBS for 1 h. Flow cytometry was performed using a FITC-conjugated goat anti-mouse IgG, with a total of 10,000 cells counted in each group. All experiments were performed at least twice under similar conditions and yielded similar results.

## Data Availability

Publicly available datasets were analyzed in this study. The genome data can be found here: SAMN03770844 and SAMN03770845, which are publicly accessible at https://www.ncbi.nlm.nih.gov/nuccore/ (accessed on 1 June 2021). The corresponding transcriptome data can be found here:CRA000357, which are publicly accessible at https://ngdc.cncb.ac.cn/gsa/browse/CRA000357 (accessed on 1 June 2021).

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
