# Peer review of "The C-Terminal Repeat Units of SpaA Mediate Adhesion of Erysipelothrix rhusiopathiae to Host Cells and Regulate Its Virulence"

_biology, 2022, doi:10.3390/biology11071010_

Round 1
Reviewer 1 Report
This manuscript began with phenotype different bacterial strains, followed by genome sequencing and comparative analysis, and ended with functional confirmation. It tells me a nice story. See bellowing for my suggestions or concerns.
1. "In addition, both rspA and sbp were split in two" in the abstract need detailed information or just delete it.
2. The adding of 120 nt to G4T10 spaA gene would be virulent as SE38 ? NO relevant descriptions in the manuscript.
3. I suggest the authors discussed the possible usage of avirulent G4T10. Could be used as vaccine ?
4. The deletion of 120 nt of SE38 spaA gene have just produced only one successful strain ? Two strains would be better to rule out the possibility of off-target effects.
5. The avirulent G4T10 was prevalent alone or with virulent SE38 ? This 120 nt would be used as marker for the detection of avirulent G4T10 ?
Author Response
1:"In addition, both rspA and sbp were split in two" in the abstract need detailed information or just delete it.
Response 1: thank you for your comment. We have deleted it.
2:The adding of 120nt to G4T10 spaA gene would be virulent as SE38?NO relevant descriptions in the manuscript.
Response 2: thank you for your questions. I am very sorry that we did not carry out the experiment of adding 120bp to SpaA gene of avirulent G4T10, however, we deleted this 120bp of the SpaA gene from virulent SE38, according to available animal test data, compared with avirulent G4T10, the virulence of virulent SE38 without this 120bp was not completely lost. Therefore, we guess that the virulence of avirulent G4T10 may increase after obtaining this 120bp, but the virulence may not reach SE38. This may be due to SpaA as a major virulence factor, but not the only one.
3:I suggest the authors discussed the possible usage of avirulent G4T10. Could be used as vaccine?
Response 3: thank you for your suggestion. We have added some discussion about the G4T10 as vaccine.
4:The deletion of 120nt of SE38 spaA gene have just produced only one successful strain? Two strains would be better to rule out the possibility of off-target effects.
Response 4: thank you for your question. We used homologous recombination to delete this 120bp of SE38, and the homologous arms in the upstream and downstream were 500bp respectively. The homologous arms were long enough to ensure good targeting. For example, Ma Z, et al. A streptococcal Fic domain-containing protein disrupts blood-brain barrier integrity by activating moesin in endothelial cells. PLoS Pathog. 2019 May 9;15(5): e1007737, D. Takamatsu, et al. Thermosensitive suicide vectors for gene replacement in Streptococcus suis, Plasmid 46 (2001) 140–148, and, Wang Y, et al. iTRAQ-based quantitative proteomic analysis reveals potential virulence factors of Erysipelothrix rhusiopathiae. J Proteomics. 2017 May 8;160:28-37. The homologous recombination method was used in all the articles above, and no off-target effects occurred, indicating that this method is widely used.
5:The avirulent G4T10 was prevalent alone or with virulent SE38?This 120nt would be used as marker for the detection of avirulent G4T10?
Response 5: I am very sorry that the data we presented were not clear enough. Both GT410 and SE38 were collected in July 2013 (accession numbers: SAMN03770844 and SAMN03770845 are described in section 2.14), so the avirulent G4T10 was prevalent with virulent SE38. We added some discussion on this 120bp would be used as marker for the detection of avirulent G4T10.

Reviewer 2 Report
Review for: The C-Terminal Repeat Units of SpaA Mediate Adhesion of Erysipelothrix rhusiopathiae to Host Cells and Regulate Its Virulence
I have reviewed the above titled manuscript for possible publication in MDPI biology. Wu et al presented a follow-up study to understand the pathogenic mechanism of E. rhusiopathiae. They utilized various omics approaches including genomics, transcriptomics (RNA-seq and qPCR) and bioinformatics to identify key genes and mutational differences associated with E. rhusiopathiae virulence compared to other strains. Importantly, they identified key repeat units at the C-terminus through which the adhesion of SpaA depends on and also determine E. rhusiopathiae virulence. Overall, their study provides a better understanding of E. rhusiopathiae pathogenesis.
This study is interesting, providing further understanding of E. rhusiopathiae pathogenesis and a suitable framework for further studies, especially cataloging gene sets and mutations for E. rhusiopathiae virulence as a genetic resource. All the methods are correctly applied, the results are clear and the interpretation is correct and adequate. Based on the results, the conclusions are supported. The results are relevant and could be applied.
The manuscript is well written. I recommend for acceptance and publication
Author Response
thank you for your comment, and the English language has been revised in the revised manuscript.